# FedDM: Iterative Distribution Matching for Communication-Efficient Federated Learning

**Yuanhao Xiong**[*]
UCLA
yhxiong@cs.ucla.edu

**Ruochen Wang**[*]
UCLA
ruocwang@ucla.edu

**Minhao Cheng**
HKUST
minhaocheng@ust.hk

**Felix Yu**
Google Research
felixyu@google.com

**Cho-Jui Hsieh**
UCLA
chohsieh@cs.ucla.edu

## Abstract

Federated learning (FL) has recently attracted increasing attention from academia and industry, with the ultimate goal of achieving collaborative training under privacy and communication constraints. Existing iterative model averaging based FL algorithms require a large number of communication rounds to obtain a well-performed model due to extremely unbalanced and non-i.i.d data partitioning among different clients. Thus, we propose FedDM to build the global training objective from multiple local surrogate functions, which enables the server to gain a more global view of the loss landscape. In detail, we construct synthetic sets of data on each client to locally match the loss landscape from original data through distribution matching. FedDM reduces communication rounds and improves model quality by transmitting more informative and smaller synthesized data compared with unwieldy model weights. We conduct extensive experiments on three image classification datasets, and results show that our method can outperform other FL counterparts in terms of efficiency and model performance. Moreover, we demonstrate that FedDM can be adapted to preserve differential privacy with Gaussian mechanism and train a better model under the same privacy budget.

## 1 Introduction

Traditional machine learning methods are designed with the assumption that all training data can be accessed from a central location. However, due to the growing data size together with the model complexity [11, 25, 31], distributed optimization [9, 12, 14] is necessary over different machines. This leads to the problem of Federated Learning [19] (FL) – multiple clients (e.g. mobile devices or local organizations) collaboratively train a global model under the orchestration of a central server (e.g. service provider) while the training data are kept decentralized and private. Such a practical setting poses two primary challenges [17, 19, 34, 42, 43]: **training data of the FL system are highly unbalanced and non-i.i.d. across downstream clients** and **more efficient communication with fewer costs is expected** because of unreliable devices with limited transmission bandwidth.

Most of the existing FL methods [19, 32, 34, 38, 50] adopt an iterative training procedure from FedAvg [19], in which each round takes the following steps: 1) The global model is synchronized with a selected subset of clients; 2) Each client trains the model locally and sends its weight or gradient back to the server; 3) The server updates the global model by aggregating messages from selected

---

[*]Equal contribution.

Workshop on Federated Learning: Recent Advances and New Challenges, in Conjunction with NeurIPS 2022 (FL-NeurIPS'22). This workshop does not have official proceedings and this paper is non-archival.

clients. This framework works effectively for generic distributed optimization while the difficult and challenging setting of FL, unbalanced data partition in particular, would result in statistical heterogeneity in the whole system [33] and make the gradient from each client inconsistent. It poses a great challenge to the training of the shared model, which requires a substantial number of communication rounds to converge [50]. Although some improvements have been made over FedAvg [19] including modifying loss functions [34], correcting client-shift with control variates [32] and the like, the reduced number of communication round is still considerable and even the amount of information required by the server rises [40].

In our paper, we propose a different iterative surrogate minimization based method, **FedDM**, referred to **Fed**erated Learning with iterative **D**istribution **M**atching. Instead of the commonly-used scheme where each client maintains a locally trained model respectively and sends its gradient/weight to the server for aggregation, we take a distinct perspective at the client's side and attempt to build a local surrogate function to approximate the local training objective. By sending those local surrogate functions to the server, the server can then build a global surrogate function around the current solution and conduct the update by minimizing this surrogate. The question is then how to build local surrogate functions that are informative and with a relative succinct representation. Inspired by recent progresses in data condensation [39, 49] we build local surrogate functions by learning a synthetic dataset to replace the original one to approximate the objective. It can be achieved by matching the original data distribution in the embedding space with the maximum mean discrepancy measurement (MMD) [10]. After the optimization of synthesized data, the client can transmit them to the server, which can then leverage the synthetic dataset to recover the global objective function for training. Our method enables the server to have implicit access to the global objective defined by the whole balanced dataset from all clients, and thus outperforms previous algorithms involved in training a local model with unbalanced data in terms of communication efficiency and effectiveness. We also demonstrate that our method can be adapted to preserve differential privacy under a modest budget, an important factor to the deployment of FL systems.

Our contributions are primarily summarized as follows:

- We propose FedDM, which is based on iterative distribution matching to learn a surrogate function. It sends synthesized data to the server rather than commonly-used local model updates and improves communication efficiency and effectiveness significantly.

- We further analyze how to protect privacy of client's data for our method and show that it is able to guarantee $(\epsilon, \delta)$-differential privacy with the Gaussian mechanism and train a better model under the same privacy budget.

- We conduct comprehensive experiments on three tasks and demonstrate that FedDM is better than its FL counterparts in communication efficiency as well as the final model performance.

## 2 Methodology

In this part, we first present the iterative surrogate minimization framework in Section 2.1, and then expand on the details of our implementation of FedDM in Section 2.2. In addition, we discuss preserving differential privacy of our method through Gaussian mechanism in Section D.

### 2.1 Iterative surrogate minimization framework for federated learning

Neural network training can be formulated as solving the following finite sum minimization problem:

$$\min_{w} f(\mathcal{D}; w) \quad \text{where} \quad f(\mathcal{D}; w) = \frac{1}{n} \sum_{i=1}^{n} \ell(x_i, y_i; w), \tag{1}$$

where $w \in \mathbb{R}^d$ is the parameter to be optimized, $\mathcal{D}$ is the dataset and $\ell(x_i, y_i; w)$ is the loss of the prediction on sample $(x_i, y_i) \in \mathcal{D}$ w.r.t. $w$ such as cross entropy. We will abbreviate these terms as $f(w)$ and $\ell_i(w)$ for simplicity. Eq. equation 1 is typically solved by stochastic optimizers when training data are gathered in a single machine. However, the scenario is different under the setting of federated learning with $K$ clients. In detail, each client $k$ has access to its local dataset of the size $n_k$

with the set of indices $\mathcal{I}_k$ ($n_k = |\mathcal{I}_k|$), and we can rewrite the objective as

$$f(w) = \sum_{k=1}^{K} \frac{n_k}{n} f_k(w) \quad \text{where} \quad f_k(w) = \frac{1}{n_k} \sum_{i \in \mathcal{I}_k} l_i(w). \tag{2}$$

Since information can only be communicated between the server and clients, previous methods [19, 32, 34, 38] train the global model by aggregation of local model updates, as introduced in Section 1. However, as each client only sees local data which could be biased and skewed, the local updates is often insufficient to capture the global information. Further, since local weight update consists limited information, it is hard for the server to obtain better joint update direction by considering higher order interactions between different clients. We are motivated to leverage the surrogate function by the example in Figure 1. Specifically, we synthesize a 1-D binary classification problem and learn a surrogate for the objective function. We learn the surrogate function via

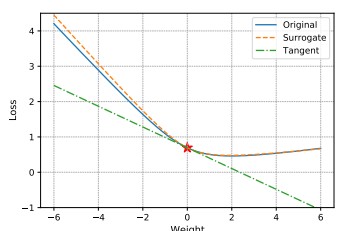

Figure 1: A 1-D example showing advantages of the surrogate function.

distribution matching introduced in Section 2.2 around the weight of 0. Compared with the tangent line computed by the gradient, the surrogate function in orange matches the original one accurately and minimizing it leads to a satisfactory solution. More details can be checked in Appendix C. Thus, we hope to develop a novel scheme such that each client can send a **local surrogate function** instead of a single gradient or weight update to the server, so the server has a more global view to loss landscape to obtain a better update instead of pure averaging.

To achieve this goal, we propose to conduct federated learning with an iterative surrogate minimization framework. At each communication round, let $w_r$ be the current solution, we build a surrogate training objective $\hat{f}_r(\cdot)$ to approximate the original training objective in the local area around $w_r$, and then update the model by minimizing the local surrogate function. The update rule can be written as

$$w_{r+1} = \min_{w \in B_\rho(w_r)} \hat{f}_r(w), \text{ where } \hat{f}_r(w) \approx f(w), \ \forall w \in B_\rho(w_r). \tag{3}$$

$B_\rho(w_r)$ is a $\rho$-radius ball around $w_r$. Note that we do not expect to build a good surrogate function in the entire parameter space; instead, we only construct it near $w_r$ and obtain the update by minimizing the surrogate function within this space. In fact, many optimization algorithms can be described under this framework. For instance, if $\hat{f}_r(w) = \nabla f(w_r)^T (w - w_r)$ (based on the first-order Taylor expansion), then Eq. equation 3 leads to the gradient descent update where $\rho$ controls the step size.

To apply this framework in the federated learning setting, we consider the decomposition of Eq. equation 2 and try to build surrogate functions to approximate each $f_k(w)$ on each client. More specifically, each client aims to find

$$\hat{f}_{r,k}(w) \approx f_k(w), \ \forall w \in B_\rho(w_t) \tag{4}$$

and send the **local surrogate function** $\hat{f}_{r,k}(\cdot)$ instead of gradient or weights to the server. The server then form the **aggregated surrogate function**

$$\hat{f}_r(w) = \hat{f}_{r,1}(w) + \cdots + \hat{f}_{r,K}(w) \tag{5}$$

and then use Eq. equation 3 to obtain the update. Again, if each $\hat{f}_{r,k}$ is the Taylor expansion based on local data, it is sufficient for the client to send local gradient to the server, and the update will be equivalent to (large batch) gradient descent. However, we will show that there exists other ways to build local approximations to make federated learning more communication efficient.

## 2.2 Local distribution matching

Inspired by recent progresses in data distillation [35, 39, 44, 48, 49], it is possible to learn a set of synthesized data for each client to represent original data in terms of the objective function. Therefore, we propose to build local surrogate models based on the following approximation for the $r$-th round:

$$f_k(w) = \frac{1}{n_k} \sum_{i \in \mathcal{I}_k} f_i(w) \approx \frac{1}{n_k^s} \sum_{j \in \mathcal{I}_k^S} \ell_j(\tilde{x}_j, \tilde{y}_j; w) = \hat{f}_{r,k}(\mathcal{S}; w), \ \forall w \in B_\rho(w_r), \tag{6}$$

where $\mathcal{S}$ denotes the set of synthesized data and $\mathcal{I}_k^{\mathcal{S}}$ is the corresponding set of indices. Note that we aim to approximate $f_k$ only in a local region around $w_r$ instead of finding the approximation globally, which is hard as demonstrated in [39, 49]. To form the approximation function in Eq. equation 6, we solve the following minimization problem:

$$\min_{\mathcal{S}} \mathbb{E}_{w \sim \mathcal{P}_w} \| f_k(w) - \hat{f}_{r,k}(\mathcal{S}; w) \|^2 \tag{7}$$

where $w$ is sampled from distribution $\mathcal{P}_w$, which is a Gaussian distribution truncated at radius $\rho$. A different perspective at Eq. equation 7 is that we can just match the distribution between the real data and synthesized ones given $f_k(w)$ and $\hat{f}_{r,k}(\mathcal{S}; w)$ are just empirical risks. A common way to achieve this is to estimate the real data distribution in the latent space with a lower dimension by maximum mean discrepancy (MMD) [49, 51]:

$$\sup_{\|h_w\|_{\mathcal{H} \leq 1}} (\mathbb{E}[h_w(\mathcal{D})] - \mathbb{E}[h_w(\mathcal{S})]). \tag{8}$$

Here $h_w$ is the embedding function that maps the input into the hidden representation. We use the empirical estimate of MMD in [49] since the underlying data distribution is inaccessible. Furthermore, to make our approximation more accurate and effective, we match the outputs of the logit layer which corresponds exactly with the Eq. equation 7, together with the preceding embedding layer:

$$\begin{aligned}
\mathcal{L} = \| \frac{1}{|\mathcal{D}|} \sum_{(x,y) \in \mathcal{D}} h_w(x) - \frac{1}{|\mathcal{S}|} \sum_{(\tilde{x},\tilde{y}) \in \mathcal{S}} h_w(\tilde{x}) \|^2 \\
+ \| \frac{1}{|\mathcal{D}|} \sum_{(x,y) \in \mathcal{D}} z_w(x) - \frac{1}{|\mathcal{S}|} \sum_{(\tilde{x},\tilde{y}) \in \mathcal{S}} z_w(\tilde{x}) \|^2,
\end{aligned} \tag{9}$$

where $h_w(x)$ again denotes intermediate features of the input while $z_w(x) \in \mathbb{R}^C$ represents the output of the final logit layer. It should be emphasized that we learn synthesized data for each class respectively, which means samples in $\mathcal{D}$ and $\mathcal{S}$ belong to the same class. For training, we adopt mini-batch based optimizers to make it more efficiently. Specifically, a batch of real data and a batch of synthetic data are sampled randomly for each class independently by $B_c^{\mathcal{D}_k} \sim \mathcal{D}_k$ and $B_c^{\mathcal{S}_k} \sim \mathcal{S}_k$. We plug these two batches into Eq. equation 9 to compute $\mathcal{L}_c$ and $\mathcal{L} = \sum_{c=0}^{C-1} \mathcal{L}_c$. $\mathcal{S}_k$ can be updated with SGD by minimizing $\mathcal{L}$ for each client.

Then we aggregate all synthesized data from $K$ clients at the server's side:

$$f(w) = \sum_{k=1}^{K} \frac{n_k}{n} f_k(w) \approx \sum_{k=1}^{K} \frac{n_k^{\mathcal{S}}}{n} \hat{f}_{r,k}(\mathcal{S}_k; w), \ \ \forall w \in B_\rho(w_r). \tag{10}$$

Moreover, since synthesized data are trained based on a specific distribution around the current value of $w$, we need to iteratively synchronize the global weights with all the clients and obtain proper $\mathcal{S}$ according to the latest $w$ for the next communication round.

Therefore, instead of transmitting parameters or gradients in previous FL algorithms, we propose federated learning with iterative distribution matching (FedDM) in Algorithm 1 following the steps below to train the global model: (a) At each communication round, for each client, we adopt Eq. equation 9 as the objective function to train synthesized data for each class; (b) The server receives synthesized data and leverages them to update the global model; (c) The current weight is then synchronized with all the clients and a new communication rounds start by repeating step (a) and (b).

It should be noticed that through estimating the local objective, FedDM extracts richer information than existing model averaging based methods, and enables the server to explore the loss landscape from a more global view. It reduces communication rounds significantly. On the other hand, the explicit message uploaded to the server, or the number of float parameters, is relatively smaller. This is especially true when training large neural network models, where the size of neural network parameters (and therefore gradient update) is much larger than the size of the input. Take CIFAR10 as an example, when training data are distributed obeying $\text{Dir}_{10}(0.5)$, the average number of classes per client (cpc) is 9. When we adopt the number of images per class (ipc) of 10 for the synthetic set, the total number of float parameters uploaded to the server is: the number of clients $\times$ cpc $\times$ ipc $\times$ image size $= 10 \times 9 \times 10 \times 3 \times 32 \times 32 \approx 2.8 \times 10^6$. For those iterative model averaging model methods, the number of float parameters is equal to the product of weight size and the number of clients, which is $320010 \times 10 \approx 3.2 \times 10^6$ for ConvNet [39] and comparably larger than FedDM. An extensive comparison is presented in Appendix E.

# 3 Experiments

## 3.1 Experimental setup

**Datasets.** In this paper, we focus on image classification tasks, and select three commonly-used datasets: MNIST [1], CIFAR10 [5], and CIFAR100 [5]. We adopt the standard training and testing split. Following commonly-used scheme [28], we simulate non-i.i.d. data partitioning with Dirichlet distribution $\text{Dir}_K(\alpha)$, where $K$ is the number of clients and $\alpha$ determines the non-i.i.d. level, and allocate divided subsets to clients respectively. A smaller value of $\alpha$ leads to more unbalanced data distribution. The default data partitioning is based on $\text{Dir}_{10}(0.5)$ with 10 clients. Furthermore, we also take into account different scenarios of data distribution, including $\text{Dir}_{10}(0.1)$, $\text{Dir}_{10}(0.01)$. Results of $\text{Dir}_{50}(0.5)$ and $\text{Dir}_{10}(50)$ (i.i.d. scenario) can be found in Appendix G.2.

**Baseline methods.** We compare FedDM with four representative iterative model averaging based methods: FedAvg [19], FedProx [34], FedNova [38], and SCAFFOLD [32]. We summarize the action of the client and the server, together with the transmitted message for all methods in Table 3.

## 3.2 Communication efficiency and convergence rate

We first evaluate our method in terms of communication efficiency and convergence rate on all three datasets on the default data partitioning $\text{Dir}_{10}(0.5)$. As we can see in Figure 2(a)-(c), our method FedDM performs the best among all considered algorithms by a large margin on MNIST, CIFAR10, and CIFAR100. Specifically, for CIFAR10, FedDM achieves $69.66 \pm 0.13\%$ on test accuracy while the best baseline SCAFFOLD only has $66.12 \pm 0.17\%$ after 20 communication rounds. FedDM also has the best convergence rate and it significantly outperforms baseline methods within the initial few rounds. Advantages of FedDM are more evident when we evaluate convergence as a function of the message size. As mentioned in 2.2, FedDM requires less information per round. Therefore, we can observe in Figure 2(d)-(e) that FedDM converges the fastest along with the message size. Details of the message size of each method for different tasks are provided in Appendix E.

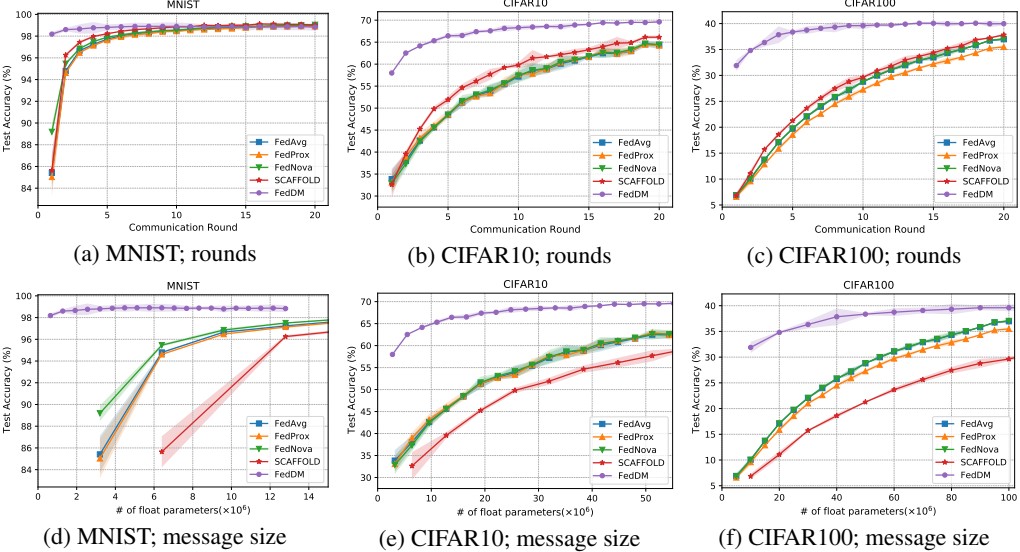

Figure 2: Test accuracy along with the number of communication rounds and the message size.

## 3.3 Evaluation on different data partitioning

In real-world applications, there are various extreme data distributions among clients. To synthesize such non-i.i.d. partitioning, we consider two more scenarios with $\text{Dir}_{10}(0.1)$ and $\text{Dir}_{10}(0.01)$. As mentioned, $\alpha \rightarrow 0$ implies each client holds examples from only one random class. It can be seen in Table 1 that previous methods based on iterative model averaging are insufficient to handle these two challenging scenarios and their performance degrades drastically compared with $\text{Dir}_{10}(0.5)$. In contrast, FedDM performs consistently better and more robustly, since distribution matching enables it to approximate the global training objective more accurately.

Table 1: Test accuracy of FL methods with different level of non-uniform data partitioning.

| Method | $\alpha = 0.1$ | | | $\alpha = 0.01$ | | |
|---|---|---|---|---|---|---|
| | MNIST | CIFAR10 | CIFAR100 | MNIST | CIFAR10 | CIFAR100 |
| FedAvg | $96.92 \pm 0.09$ | $57.32 \pm 0.04$ | $32.00 \pm 0.50$ | $91.04 \pm 0.80$ | $57.32 \pm 0.04$ | $27.05 \pm 0.45$ |
| FedProx | $96.72 \pm 0.04$ | $56.92 \pm 0.30$ | $30.77 \pm 0.52$ | $91.18 \pm 0.16$ | $40.30 \pm 0.15$ | $25.88 \pm 0.39$ |
| FedNova | $98.04 \pm 0.03$ | $60.76 \pm 0.14$ | $31.92 \pm 0.42$ | $90.27 \pm 0.49$ | $36.46 \pm 0.42$ | $27.52 \pm 0.43$ |
| SCAFFOLD | $98.32 \pm 0.06$ | $60.96 \pm 1.20$ | $34.39 \pm 0.25$ | $88.37 \pm 0.25$ | $32.42 \pm 1.13$ | $31.14 \pm 0.20$ |
| FedDM | $\mathbf{98.67 \pm 0.01}$ | $\mathbf{67.38 \pm 0.32}$ | $\mathbf{37.58 \pm 0.27}$ | $\mathbf{98.21 \pm 0.23}$ | $\mathbf{63.82 \pm 0.17}$ | $\mathbf{34.98 \pm 0.17}$ |

### 3.4 Analysis of FedDM

We analyze FedDM to investigate effects of hyperparameters such as ipc and network structure. Besides, we compare our method with a strong baseline of sending real images with the same size. More extensive results are reported in Appendix G including visualization of learned synthetic data.

**Effects of ipc.** Experiments are conducted on CIFAR10 with the distribution $\text{Dir}_{10}(0.5)$ with three different ipc values from $[3, 5, 10]$. As the ipc increases, the performance gradually get better from $53.64 \pm 0.35\%$, $62.24 \pm 0.04\%$ to $69.62 \pm 0.14\%$. However, in the meanwhile, more images per class indicates a heavier communication burden. We need to trade off the model performance against the communication cost, and thus choose an appropriate ipc value based on the task.

**Different network structures.** Besides ConvNet, we evaluate FedDM under the default CIFAR10 setting on ResNet-18. It can be observed that our method works well even for this more complicated and larger model in Figure 3. It should also be emphasized that for FL baseline methods, they have to transmit a larger amount of message while FedDM maintains the original size. This makes FedDM more efficient in larger networks.

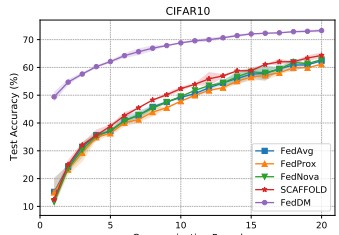

Figure 3: Test accuracy on ResNet-18.

**Comparison with transmitting real images.** Our method is compared with REAL, which sends real images of the same size as FedDM (ipc = 10). In particular, REAL achieves test accuracy of $68.66 \pm 0.08\%$ on CIFAR10 with the default setting, but cannot beat FedDM with $69.62 \pm 0.14\%$. It indicates that our learned synthetic set can capture richer information of the whole dataset rather than just a few images.

## 4 Conclusions and limitations

In this paper we propose an iterative distribution matching based method, FedDM to achieve more communication-efficient federated learning. By learning a synthetic dataset for each client to approximate the local objective function, the server can obtain the global view of the loss landscape better than just aggregating local model updates. We also show that FedDM can preserve differential privacy with Gaussian mechanism. However, there is still a trade-off between the size of the synthetic set and the final performance, especially for classification tasks with hundreds of clients or classes. How to reduce the synthetic set to save communication costs can be a potential future direction.

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

## A  Related work

**Federated Learning.**  Federated learning [19, 42] has aroused heated discussion nowadays from both research and applied areas. With the goal to train the model collaboratively, it incorporates the principles of focused data collection and minimization [42]. FedAvg [19] was proposed along with the concept of FL as the first effective method to train the global model under the coordination of multiple devices. Since it is based on iterative model averaging, FedAvg suffers from heterogeneity in the FL system, especially the non-i.i.d. data partitioning, which degrades the performance of the global model and adds to the burden of communication [33]. To mitigate the issue, some variants have been developed upon FedAvg including [32, 34, 38]. For instance, FedProx [34] modifies the loss function while FedNova [38] and SCAFFOLD [32] leverage auxiliary information to balance the distribution shift. Apart from better learning algorithms with faster convergence rate, another perspective at improving efficiency is to reduce communication costs explicitly [23, 26, 29, 36, 41]. An intuitive approach is to quantize and sparsify the uploaded weights directly [36]. Efforts have also been made towards one-shot federated learning [24, 27, 40, 46], expecting to obtain a satisfactory model through only one communication round.

**Differential Privacy.** To measure and quantify information disclosure about individuals, researchers usually adopt the state-of-the-art model, differential privacy (DP) [3, 7, 13]. DP describes the patterns of groups while withholding information about individuals in the dataset. There are many scenarios in which DP guarantee is necessary [2, 4, 15, 21, 45]. For example, Abadi et.al [15] developed differentially private SGD (DP-SGD) which enabled training deep neural networks with non-convex objectives under a certain privacy budget. It was further extended to settings of federated learning, where various techniques have been designed to guarantee user-level or local differential privacy [18, 20]. Recently, DP has been taken into consideration for hyperparameter tuning [45].

**Dataset Distillation.** With the explosive growing of the size of training data, it becomes much more challenging and costly to acquire large datasets and train a neural network within moderate time [35, 44]. Thus, constructing smaller but still informative datasets is of vital importance. The traditional way to reduce the size is through coreset selection [8, 30], which select samples based on particular heuristic criteria. However, this kind of method has to deal with a trade-off between performance and data size [35, 39]. To improve the expressiveness of the smaller dataset, recent approaches consider learning a synthetic set of data from the original set, or data distillation for simplicity. Along this line, different methods are proposed using meta-learning [22, 47], gradient matching [39, 48], distribution matching [49, 51], neural kernels [35, 44] or generative models [37].

## B FedDM Algorithm

---

**Algorithm 1:** FedDM: Federated Learing with Distribution Matching

---

1  **Input:** Training set $\mathcal{D}$, set of synthetic samples $\mathcal{S}$, deep neural network parameterized with $w$, probability distribution over parameters $\mathcal{P}_w$, gradient norm bound $\mathcal{C}$, training iterations of distribution matching $T$, learning rate $\eta_c$ and $\eta_s$.

2  **Server executes:**

3  **for** *each round* $r = 1, \ldots, R$ **do**

4     **for** *client* $k = 1, \ldots, K$ **do**

5         $\mathcal{S}_k \leftarrow$ ClientUpdate$(k, w_r)$

6         Transmit $\mathcal{S}_k$ to the server

7     Aggregate synthesized data from each client and build the surrogate function by Eq. equation 10

8     Update weights to $w_{r+1}$ on $\mathcal{S}$ by SGD with the learning rate $\eta_s$

9  **ClientUpdate**$(k, w_r)$:

10  Initialize $\mathcal{S}_k$ from random noise or real examples.

11  **for** $t = 0, \cdots, T - 1$ **do**

12     Sample $w \sim P_w(w_r)$

13     Sample mini-batch pairs $B_c^{\mathcal{D}_k} \sim \mathcal{D}_k$ and $B_c^{\mathcal{S}_k} \sim \mathcal{S}_k$ for each class $c$

14     Compute $\mathcal{L}_c$ based on Eq. equation 9, $\mathcal{L} \leftarrow \sum_{c=0}^{C-1} \mathcal{L}_c$

15     Update $\mathcal{S}_k \leftarrow \mathcal{S}_k - \eta_c \nabla_{\mathcal{S}_k} \mathcal{L}$

---

## C Synthetic Binary Classification

We design a synthetic 1-D binary classification problem to better illustrate the advantage of learning a surrogate function for the training objective. Specifically, we construct a dataset $\mathcal{D}_s = \{(x_i, y_i) | i = 1, \ldots, n\}$ with $n = 100$ synthetic pairs in the following way:

$$x_i \sim \mathcal{N}(0, 1), y_i = \begin{cases} 1 & (x_i \geq 0 \text{ and } p_i \geq 0.1) \text{ or } (x_i < 0 \text{ and } p_i < 0.1) \\ 0 & \text{otherwise} \end{cases}, \tag{11}$$

where $p_i$ is a random value sampled from Uniform$(0, 1)$. A prediction is made by $\hat{y}_i = \text{Sigmoid}(wx_i)$ with the weight $w$ as the trainable parameter. We use the binary cross entropy as the training objective:

$$\mathcal{L}_{\text{BCE}} = -\frac{1}{n} \sum_{i=1}^{n} y_i \log(\hat{y}_i) + (1 - y_i) \log((1 - \hat{y}_i)). \tag{12}$$

Then we use $n' = 20$ randomly initialized examples $\{(\tilde{x}_j, \tilde{y}_j) | j = 1, \ldots, 20\}$ to match the objective around $w = 0$ as introduced in Section 2.2. We plot the original objective, the surrogate function, and the tangent line at $w = 0$ obtained by the gradient in Figure 1.

# D  Differential privacy of FedDM

An important factor to evaluate a federated learning algorithm is whether it can preserve differential privacy. Before analyzing our method, we first review some fundamentals of differential privacy.

**Definition D.1 (Differential Privacy [2]).** A randomized mechanism $\mathcal{M} : \mathcal{D} \rightarrow \mathcal{R}$ with domain $\mathcal{D}$ and range $\mathcal{R}$ satisfies $(\epsilon, \delta)$-differential privacy if for any two adjacent datasets $D_1, D_2$ and any measurable subset $S \subseteq \mathcal{R}$,

$$\Pr(\mathcal{M}(D_1) \in S) \leq e^\epsilon \Pr(\mathcal{M}(D_2) \in S) + \delta. \tag{13}$$

In this paper, we focus on instance-level differential privacy, which indicates that $D_1$ and $D_2$ differ on a single element. Typically, the randomized mechanism is applied to a query function of the dataset, $f : \mathcal{D} \rightarrow \mathcal{X}$. Without loss of generality, we assume that the output spaces $\mathcal{R}, \mathcal{X} \subseteq \mathbb{R}^m$. A key quantity in characterizing differential privacy for various mechanisms is the sensitivity of a query [13] $f : \mathcal{D} \rightarrow \mathbb{R}^m$ in a given norm $\ell_p$. Formally this is defined as

$$\Delta_p \triangleq \max_{D_1, D_2} \|f(D_1) - f(D_2)\|_p. \tag{14}$$

Gaussian mechanism [13] is one simple and effective method to achieve $(\epsilon, \delta)$-differential privacy:

$$\mathcal{M}(D) \triangleq f(D) + Z, \quad \text{where} \quad Z \sim \mathcal{N}(0, \sigma^2 \Delta_p^2 \mathbf{I}). \tag{15}$$

It has been proved that under Gaussian mechanism, $(\epsilon, \delta)$-differential privacy is satisfied for the function $f$ of sensitivity $\Delta_p$ if we choose $\sigma \geq \sqrt{2 \log \frac{1.25}{\delta}} / \epsilon$ [13]. Differentially private SGD (DP-SGD) [15] then applies Gaussian mechanism to deep learning optimization with hundreds of steps and demonstrates the following theorem:

**Theorem D.1 (Differential Privacy of DP-SGD).** There exist constants $c_1$ and $c_2$ so that given the sampling probability $q$ and the number of steps $T$, for any $\epsilon < c_1 q^2 T$, DP-SGD is $(\epsilon, \delta)$-differentially private for any $\delta > 0$ if

$$\sigma \geq c_2 \frac{q\sqrt{T \log(1/\delta)}}{\epsilon}. \tag{16}$$

We then prove that by leveraging DP-SGD to update the synthetic dataset which is initialized from random Gaussian noise, FedDM can preserve differential privacy of the original dataset. We present this DP guarantee of FedDM in the theorem below:

**Theorem D.2 (Differential privacy of FedDM.).** Given the synthetic dataset $\mathcal{S}$ is initialized from random noise, FedDM trained with DP-SGD can guarantee $(\epsilon, \delta)$-differential privacy in a K-client federated learning system, with $\sigma \geq \sqrt{\frac{\log(\delta)}{Tq^2 - \epsilon}}$ or $\sigma \geq \sqrt{\frac{2\log(1/\sigma)}{\epsilon}}$ if $Tq^2 \leq \epsilon/2$ in each round.

The complete proof of Theorem D.2 is presented below.

According to Theorem D.1, using DP-SGD to train a neural network can protect differential privacy. Back to FedDM, we first look at each client separately to investigate its differential privacy. We show that the gradient of $\mathcal{L}_c$ in equation 9 can be written as the average of individual gradients for each real example,

$$\nabla_{\mathcal{S}_k} \mathcal{L}_c = \frac{1}{|B_c^{\mathcal{D}_k}|} \sum_{(x_i, y_i) \in B_c^{\mathcal{D}_k}} \tilde{g}(x_i), \tag{17}$$

where $\tilde{g}(x_i)$ is the modified gradient for $x_i$. Recall the equation of $\mathcal{L}_c$, we have

$$\mathcal{L}_c = \overbrace{\|\frac{1}{|B_c^{\mathcal{D}_k}|}\sum_{(\boldsymbol{x},y)\in B_c^{\mathcal{D}_k}} h_w(x) - \frac{1}{|B_c^{\mathcal{S}_k}|}\sum_{(\tilde{x},\tilde{y})\in B_c^{\mathcal{S}_k}} h_w(\tilde{x})\|^2}^{\mathcal{L}_{c,h}}$$
$$+ \underbrace{\|\frac{1}{|B_c^{\mathcal{D}_k}|}\sum_{(x,y)\in B_c^{\mathcal{D}_k}} z_w(x) - \frac{1}{|B_c^{\mathcal{S}_k}|}\sum_{(\tilde{x},\tilde{y})\in B_c^{\mathcal{S}_k}} z_w(\tilde{x})\|^2}_{\mathcal{L}_{c,z}}. \tag{18}$$

$\mathcal{L}_c$ are divided into two similar parts, $\mathcal{L}_{c,h}$ and $\mathcal{L}_{c,z}$. Then we take a look at the gradient of $\mathcal{L}_{c,h}$ with respect to $\mathcal{S}_k$ below:

$$\nabla_{\mathcal{S}_k}\mathcal{L}_{c,h} = 2(\overbrace{\frac{\partial\frac{1}{|B_c^{\mathcal{S}_k}|}\sum_{(\tilde{x},\tilde{y})\in B_c^{\mathcal{S}_k}} h_w(\tilde{x})}{\partial \mathcal{S}_k}}^{J_{\mathcal{S}_k}})^T(\overbrace{\frac{1}{|B_c^{\mathcal{S}_k}|}\sum_{(\tilde{x},\tilde{y})\in B_c^{\mathcal{S}_k}} h_w(\tilde{x})}^{h_w(\mathcal{S}_k)} - \frac{1}{|B_c^{\mathcal{D}_k}|}\sum_{(x,y)\in B_c^{\mathcal{D}_k}} h_w(x))$$

$$= J_{\mathcal{S}_k}(h_w(\mathcal{S}_k) - \frac{1}{|B_c^{\mathcal{D}_k}|}\sum_{(x,y)\in B_c^{\mathcal{D}_k}} h_w(x))$$

$$= \frac{1}{|B_c^{\mathcal{D}_k}|}\sum_{(x,y)\in B_c^{\mathcal{D}_k}} \underbrace{J_{\mathcal{S}_k}(h_w(\mathcal{S}_k) - h_w(x))}_{\tilde{h}_w(x)} = \frac{1}{|B_c^{\mathcal{D}_k}|}\sum_{(x,y)\in B_c^{\mathcal{D}_k}} \tilde{h}_w(x). \tag{19}$$

Similarly, we have

$$\nabla_{\mathcal{S}_k}\mathcal{L}_{c,z} = \frac{1}{|B_c^{\mathcal{D}_k}|}\sum_{(x,y)\in B_c^{\mathcal{D}_k}} \tilde{z}_w(x). \tag{20}$$

Then the final gradient of $\mathcal{L}_c$ is

$$\nabla_{\mathcal{S}_k}\mathcal{L}_c = \nabla_{\mathcal{S}_k}\mathcal{L}_{c,h} + \nabla_{\mathcal{S}_k}\mathcal{L}_{c,z} = \frac{1}{|B_c^{\mathcal{D}_k}|}\sum_{(x,y)\in B_c^{\mathcal{D}_k}} (\overbrace{\tilde{h}_w(x) + \tilde{z}_w(x)}^{\tilde{g}(x)}) = \frac{1}{|B_c^{\mathcal{D}_k}|}\sum_{(x,y)\in B_c^{\mathcal{D}_k}} \tilde{g}(x). \tag{21}$$

It indicates that the synthetic set can be regarded as an equivalence to the network parameter in DP-SGD, and leads to the conclusion that for each client $k$, Theorem D.1 holds during optimizing $\mathcal{S}_k$. Furthermore, since the synthetic dataset is initialized from random noise, it would not leak privacy at the beginning of the optimization.

Next, to extend DP guarantee to a system with $K$ clients, we use parallel composition [6]:

**Theorem D.3 (Parallel Composition [6]).** If there are $K$ mechanisms $M_1, \ldots, M_K$ computed on disjoint subsets whose privacy guarantees are $(\epsilon_1, \delta_1), \ldots, (\epsilon_K, \delta_K)$ respectively, then any function of $M_1, \ldots, M_K$ is $(\max_i \epsilon_i, \max_i \delta_i)$-differential private.

We can see that different clients maintain their own local datasets, which satisfies disjoint property. Then this Gaussian mechanism is still $(\epsilon, \delta)$-differentially private for the whole system if each client satisfies $(\epsilon, \delta)$-differential privacy. In addition, to quantify how much noise is required for each client, we can make use of the Tail bound in [15]:

$$\delta = \min_{\lambda} \exp(\alpha_M(\lambda) - \lambda\epsilon). \tag{22}$$

Based on [15], $\alpha_M(\lambda) \leq Tq^2\lambda^2/\sigma^2$, without loss of generality, set $\lambda = \sigma^2$, it holds that $\delta \leq \exp(Tq^2\sigma^2 - \epsilon\sigma^2)$, and $\sigma \geq \sqrt{\frac{\log(\delta)}{Tq^2-\epsilon}}$. When $Tq^2 \leq \epsilon/2$, we have $\sigma \geq \sqrt{\frac{2\log(1/\sigma)}{\epsilon}}$.

To guarantee differential privacy when leveraging the synthetic dataset for downstream tasks, we introduce the property of post-processing below:

**Lemma D.1 (Robustness to post-processing [13]).** Let $\mathcal{M} : \mathcal{D} \to \mathcal{R}$ be a randomized mechanism that is $(\epsilon, \delta)$-differentially private. If $\mathcal{F} : \mathcal{R} \to \mathcal{R}'$ be an arbitrary deterministic or randomized mapping, $\mathcal{F}(\mathcal{M})$ is also $(\epsilon, \delta)$-differentially private.

Training a network on the synthetic dataset $\mathcal{S}$ is a post-processing operation, and Lemma D.1 ensures that any post-processing computation on $\mathcal{S}$ is differentially private as long as the generation of $\mathcal{S}$ satisfies $(\epsilon, \delta)$-differential privacy. This property has also been deployed in similar applications such as [52], where the synthetic text generated from a DP-trained language model was used in various downstream tasks and still preserved privacy.

Finally, with Theorem D.1, Theorem D.3 and Lemma D.1, we complete the proof of Theorem D.2. The whole procedure of FedDM integrated with DP-SGD is described in Algorithm 2.

---

**Algorithm 2:** FedDM with DP-SGD

---

1   **Input:** Training set $\mathcal{D}$, set of synthetic samples $\mathcal{S}$, deep neural network parameterized with $w$, probability distribution over parameters $\mathcal{P}_w$, Gaussian noise level $\sigma$, gradient norm bound $\mathcal{C}$, training iterations of distribution matching $T$, learning rate $\eta_c$ and $\eta_s$.

2   **Server executes:**

3   **for** *each round* $r = 1, \ldots, R$ **do**

4      **for** *client* $k = 1, \ldots, K$ **do**

5          $\mathcal{S}_k \leftarrow \text{ClientUpdate}(k, w_r, \sigma)$

6          Transmit $\mathcal{S}_k$ to the server

7      Aggregate synthesized data from each client and build the surrogate function by Eq. equation 10

8      Update weights to $w_{r+1}$ on $\mathcal{S}$ by SGD with the learning rate $\eta_s$

9   **ClientUpdate**$(k, w_r, \sigma)$:

10   Initialize $\mathcal{S}_k$ with random noise.

11   **for** $t = 0, \cdots, T - 1$ **do**

12      Sample $w \sim P_w(w_r)$

13      Sample mini-batch pairs $B_c^{\mathcal{D}_k} \sim \mathcal{D}_k$ and $B_c^{\mathcal{S}_k} \sim \mathcal{S}_k$ for each class $c$

14      Compute $\mathcal{L}_c$ based on Eq. equation 9, $\mathcal{L} \leftarrow \sum_{c=0}^{C-1} \mathcal{L}_c$

15      Obtain the clipped gradient: $\nabla_{\mathcal{S}_k} \mathcal{L}_c \leftarrow \nabla_{\mathcal{S}_k} \mathcal{L}_c / \max\left(1, \frac{\|\nabla_{\mathcal{S}_k} \mathcal{L}_c\|_2}{\mathcal{C}}\right)$

16      Add Gaussian noise: $\nabla_{\mathcal{S}_k} \mathcal{L}_c \leftarrow \nabla_{\mathcal{S}_k} \mathcal{L}_c + \frac{1}{|B_c^{\mathcal{D}_k}|} \mathcal{N}(0, \sigma^2 \mathcal{C}^2 \mathbf{I})$

17      Update $\mathcal{S}_k \leftarrow \mathcal{S}_k - \eta_c \nabla_{\mathcal{S}_k} \mathcal{L}$

---

## E   Message size of different FL methods

In this section, we provide specific message size under different data partitioning of FedDM. As discussed previously, the message size of all baseline methods are determined on the model size, while the message size varies from different scenarios and ipc values. When there are 10 clients, we set ipc=10 for MNIST and CIFAR10, and ipc=5 for CIFAR100. We present the results in Table 2. It can be observed that FedDM are more advantageous for unbalanced data partitioning, such as $\text{Dir}_{10}(0.1)$ and $\text{Dir}_{10}(0.01)$. For the experiment of $\text{Dir}_{50}(0.5)$ on CIFAR10 with ConvNet, the message sizes of FedDM and baselines are $3.1 \times 10^6$ and $1.6 \times 10^7$ respectively, where our method saves about 80% costs per round. Moreover, if the underlying model are changed to ResNet-18 for $\text{Dir}_{10}(0.5)$, then the number of parameters is about $1.1 \times 10^8$.

## F   Implementation Details

**Hyperparameters.** For FedDM, following [49], we select the batch size as 256 for real images, and update the synthetic set $\mathcal{S}_k$ for $T = 1,000$ iterations with $\eta_c = 1$ for each client in each communication round, and tune the number of images per class (ipc) within the range $[3, 5, 10]$. Synthetic images are initialized as randomly sampled real images with corresponding labels suggested by [39, 49]. Considering the trade-off between communication efficiency and model performance,

Table 2: The size of message uploaded to the server (number of float parameters).

|  | MNIST | CIFAR10 | CIFAR100 |
|---|---|---|---|
| $\mathrm{Dir}_{10}(0.5)$ | 635040 | 2672640 | 10045440 |
| $\mathrm{Dir}_{10}(0.1)$ | 368480 | 1351680 | 4761600 |
| $\mathrm{Dir}_{10}(0.01)$ | 109760 | 460800 | 2135040 |
| Baseline | 3177060 | 3200100 | 5044200 |

Table 3: Summary of different FL methods.

| Method | Client | Message | Server |
|---|---|---|---|
| FedAvg [19] | $\min f_k(w)$ | $\Delta_w$* | model averaging |
| FedProx [34] | $\min f_k(w) + \mu\|w - w_r\|/2$ | $\Delta_w$ | model averaging |
| FedNova [38] | $\min f_k(w)$ | $d$ and $a$* | normalized model averaging |
| SCAFFOLD [32] | $\min f_k(w, c)$ | $\Delta_w$ and $\Delta_c$* | model averaging for both $w$ and $c$ |
| FedDM(Ours) | min Eq. equation 9 | $\mathcal{S}$ | model updating on $\mathcal{S}$ |

* $\Delta_w$ denotes the model update, $d$ is the aggregated gradient and $a$ is the coefficient vector, $\Delta_c$ is the change of control variates. Refer to original papers for more details.

we choose ipc to be 10 for MNIST and CIFAR10, 5 for CIFAR100 when there are 10 clients. The choice of radius $\rho = 5$ is discussed in Appendix G.4. On the server's side, the global model is trained with the batch size 256 for 500 epochs by SGD of $\eta_s = 0.01$. For baseline methods[2], we choose the same batch size of 256 for local training, and tune the learning rate of SGD from $[0.001, 0.01, 0.1]$ and local epoch from $[5, 10, 15, 20]$. In particular, we tune $\mu$ for FedProx in $[0.01, 0.1, 1]$. For a fair comparison, all methods share the fixed number of communication rounds as 20, and the same model structure ConvNet [39] by default. A different network ResNet-18 [16] is evaluated as well. All experiments are run for three times with different random seeds with one NVIDIA 2080Ti GPU and the average performance is reported in the paper.

# G Additional Experimental Results

## G.1 Performance with DP guarantee

As discussed in Section D, if the synthetic dataset is initialized from random noise, using DP-SGD in local training of FedDM can satisfy $(\epsilon, \delta)$-differential privacy, with $\sigma \geq \sqrt{\frac{2\log(1/\delta)}{\epsilon}}$ for any $Tq^2 \leq \epsilon/2$. Such a mechanism also works on baseline methods with the same DP guarantee. Therefore, our evaluation scheme just adopts the same level of Gaussian noise in DP-SGD given the specific budget $(\epsilon, \delta)$ and then compares performance of different algorithms. Specifically, we follow FedDM with DP-SGD in Algorithm 2 with three noise levels from small ($\sigma = 1$), medium ($\sigma = 3$), to large ($\sigma = 5$), and the gradient norm bound $\mathcal{C} = 5$. $\mathcal{S}$ is initialized from $\mathcal{N}(0, 1)$ to guarantee differential privacy. We notice in Figure 4 that under the same DP guarantee, FedDM outperforms other FL counterparts in terms of convergence rate and final performance. Moreover, the accuracy of FedDM does not drop significantly compared with all considered methods when the noise level increases from $\sigma = 1$ to $\sigma = 5$, indicating that FedDM is most resistant to the perturbed optimization.

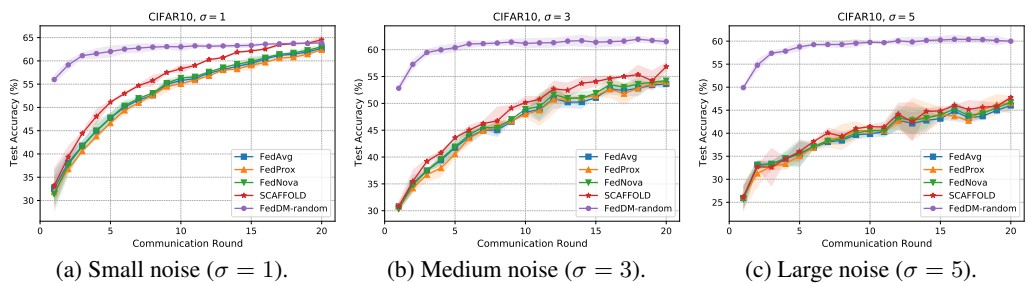

(a) Small noise ($\sigma = 1$).     (b) Medium noise ($\sigma = 3$).     (c) Large noise ($\sigma = 5$).

Figure 4: Performance of FL methods with different levels of noise.

[2]We use implementations from `https://github.com/Xtra-Computing/NIID-Bench` in [50].

## G.2 Learning curves of FL methods

We show a complete set of learning curves for all of our experiments.

**Different data partitioning.** Here we present curves for different data partitioning. We observe that FedDM still outperforms all other baselines under scenarios of $\text{Dir}_{10}(50)$ in Figure 8 which is almost an i.i.d. data partitioning, and $\text{Dir}_{50}(0.5)$ in Figure 9 which has more clients.

- $\text{Dir}_{10}(0.5)$

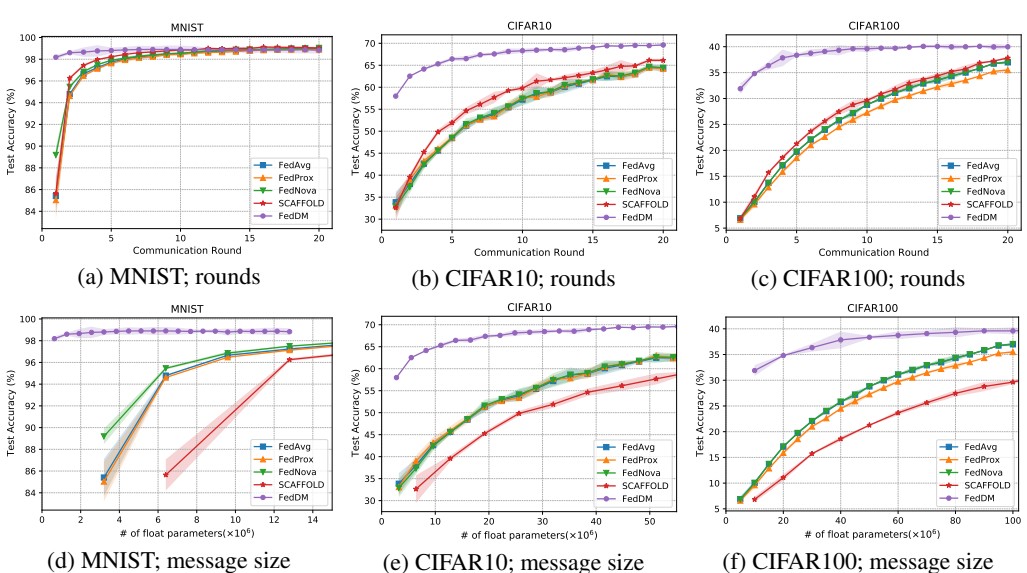

Figure 5: Test accuracy under $\text{Dir}_{10}(0.5)$.

- $\text{Dir}_{10}(0.1)$

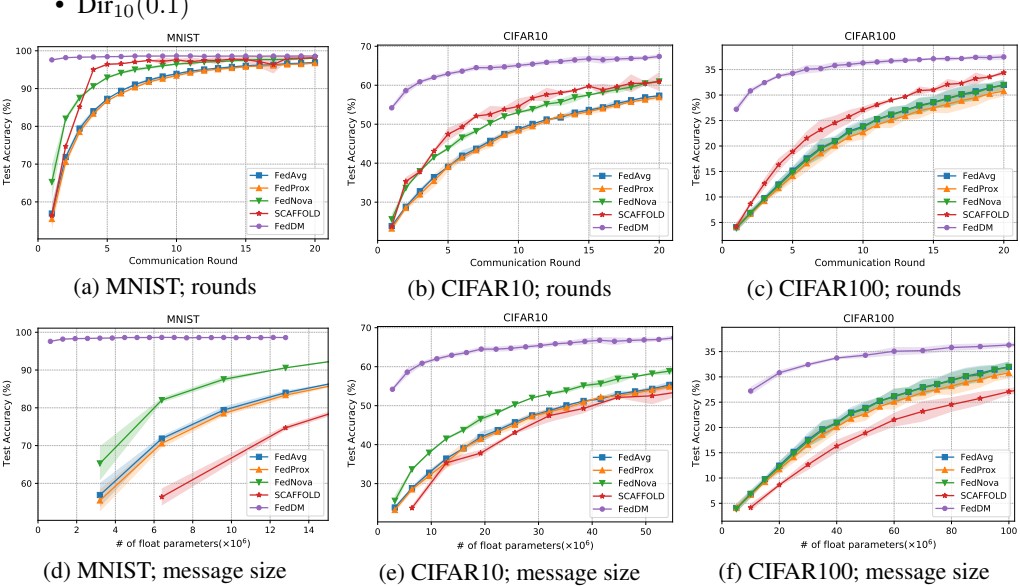

Figure 6: Test accuracy under $\text{Dir}_{10}(0.1)$.

- $\text{Dir}_{10}(0.01)$

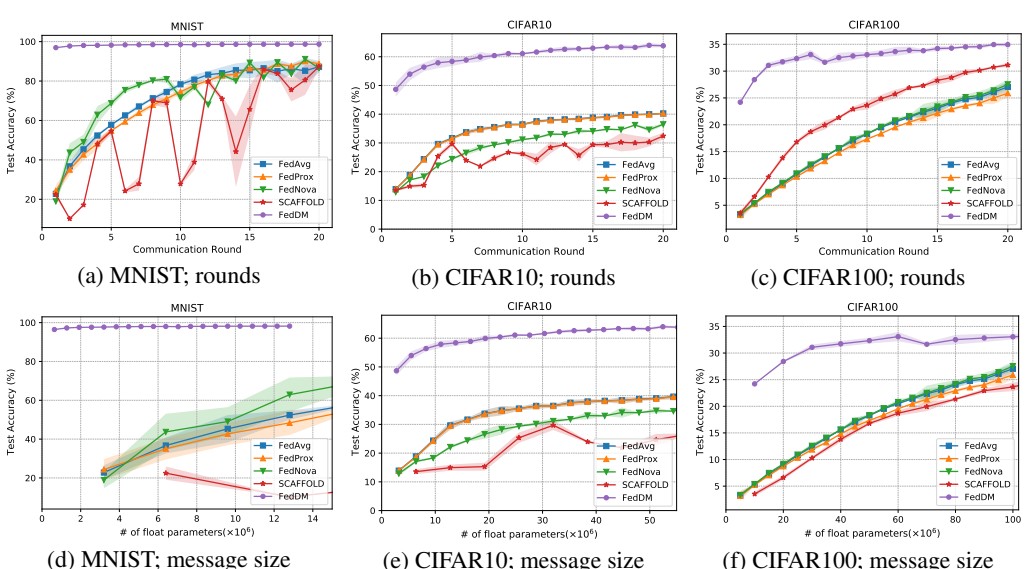

(a) MNIST; rounds  (b) CIFAR10; rounds  (c) CIFAR100; rounds

(d) MNIST; message size  (e) CIFAR10; message size  (f) CIFAR100; message size

Figure 7: Test accuracy under $\text{Dir}_{10}(0.01)$.

- i.i.d., $\text{Dir}_{10}(50)$

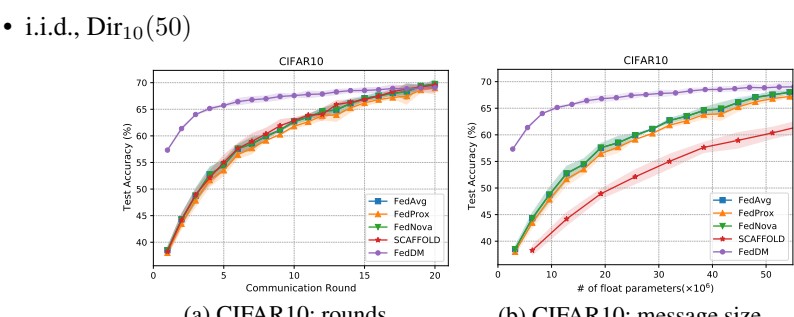

(a) CIFAR10; rounds  (b) CIFAR10; message size

Figure 8: Test accuracy under $\text{Dir}_{10}(50)$.

- $\text{Dir}_{50}(0.5)$

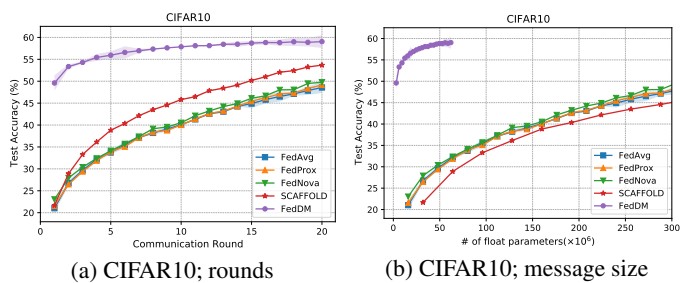

(a) CIFAR10; rounds  (b) CIFAR10; message size

Figure 9: Test accuracy under $\text{Dir}_{50}(0.5)$.

**Different noise levels.** Figure 10 displays learning curves of different $\sigma$.

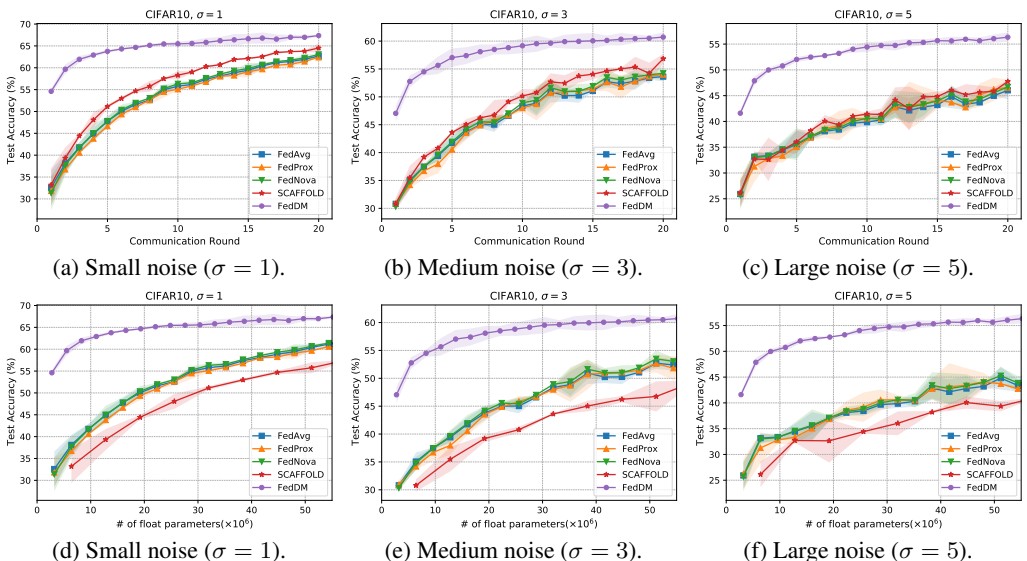

Figure 10: Performance of FL methods with different levels of noise.

**Effects of ipc.** We show test accuracy curves to analyze effects of ipc in Figure 11.

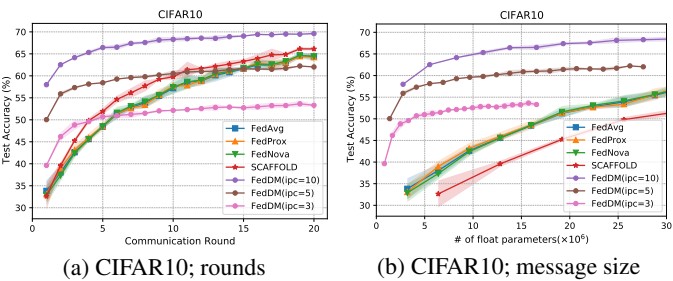

Figure 11: Performance of FedDM with different values of ipc.

**Performance on ResNet-18.** Detailed learning curves of test accuracy along with rounds and message size are shown in Figure 12.

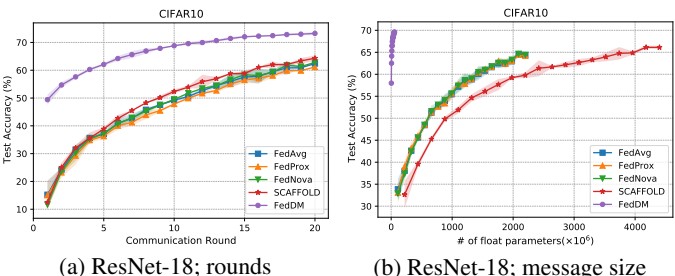

Figure 12: Performance of FL methods on ResNet-18.

**Transmitting real data.** We present a comparison with sending real images (REAL) in Figure 13.

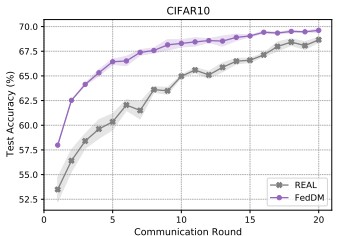

Figure 13: Test accuracy of FedDM and REAL.

### G.3 Visualization of the synthetic dataset

By randomly picking a client under data partitioning of $\mathrm{Dir}_{10}(0.5)$, we provide the visualization of our synthetic dataset under different noise levels in Figure 14, 15, 16, and 17. It can be observed clearly that even when there is no noise added to the gradient during optimization of the synthetic dataset, those images are still illegible from their original classes in Figure 14. Furthermore, as $\sigma$ increases, synthesized data become harder to recognize, which protects the client's privacy successfully.

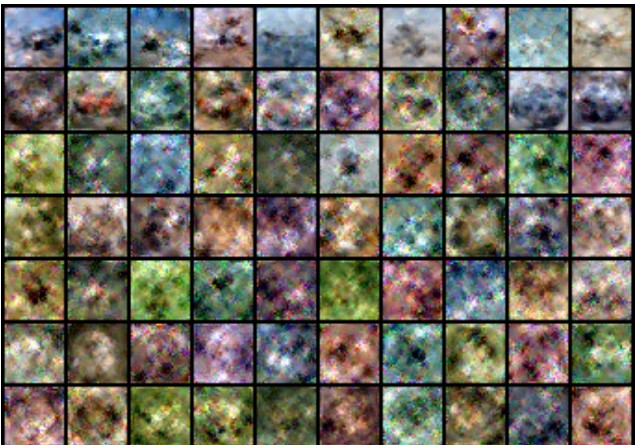

Figure 14: Synthesized images when no noise is added.

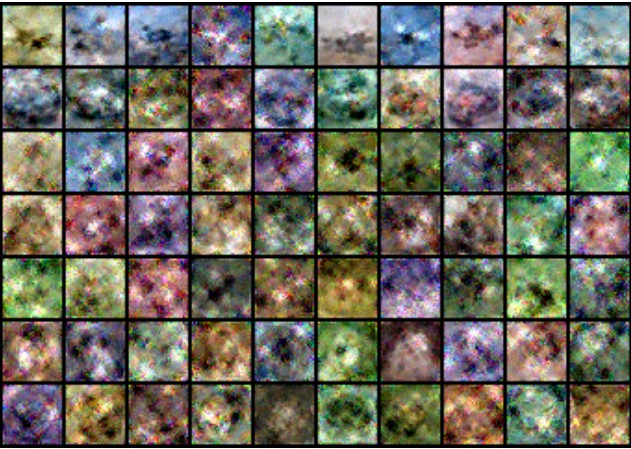

Figure 15: Synthesized images with $\sigma = 2$

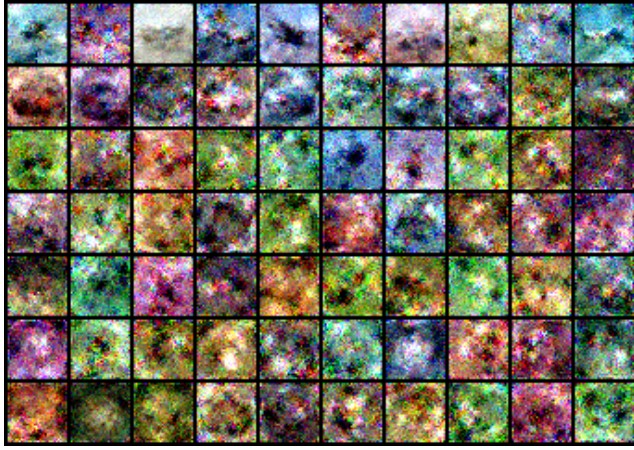

Figure 16: Synthesized images with $\sigma = 3$.

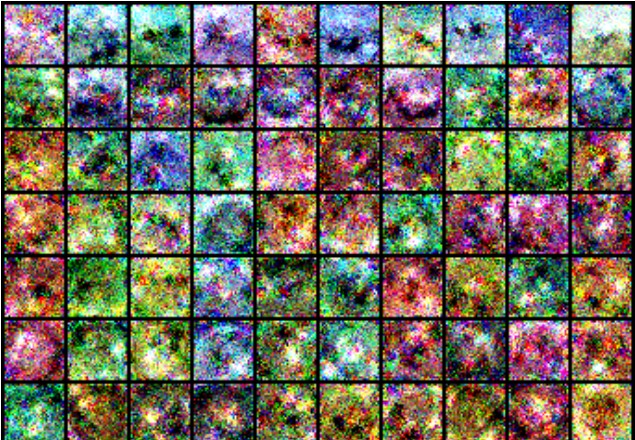

Figure 17: Synthesized images with $\sigma = 5$.

### G.4 $\rho$-radius ball

It has been discussed in Section 2.1 that $B_\rho(w_r)$ is a $\rho$-radius ball around $w_r$. Specifically,

$$B_\rho(w_r) = \{w | \|w - w_r\|_2 \le \rho\}. \tag{23}$$

In FedDM, we sample $w$ based on a truncated Gaussian distribution below:

$$P_w(w_r) = \text{Clip}(\mathcal{N}(w_r, 1), \rho), \tag{24}$$

where we clip the sampled weight to guarantee that $\|w - w_r\|_2 \le \rho$. At the server's side, when training the global model, we also clip the weight to the $\rho$-radius ball. We conduct experiments to choose $\rho$ from $[3, 5, 10]$ and present the test accuracy after 20 communication rounds on CIFAR10 under the default $\text{Dir}_{10}(0.5)$ setting in Table 4. We find that performance is similar and FedDM is not very sensitive to the choice of $\rho$. $\rho = 5$ performs relatively best and we hypothesize that too small weight makes the optimization of the global model restricted and too big one adds to the difficulty of learning a surrogate function. Based on results in Table 4, we select $\rho = 5$ for all our experiments.

Table 4: Test accuracy of FedDM under different $\rho$.

| $\rho$ | Test accuracy |
|---|---|
| $\rho = 3$ | $69.15 \pm 0.09\%$ |
| $\rho = 5$ | $69.66 \pm 0.13\%$ |
| $\rho = 10$ | $69.32 \pm 0.24\%$ |

