# OpenReview forum: "FedDM: Iterative Distribution Matching for Communication-Efficient Federated Learning"
_NeurIPS.cc/2022/Workshop/Federated_Learning — FL-NeurIPS 2022 Poster_

### Official Review · Reviewer_kLur · 2022-10-15
**nice paper on improving convergence of FL via synthetic data generation**

This is an interesting paper, trying to improve the convergence speed of Federated Learning from synthetic data generation perspective, rather than pure optimization angle.

The paper is well-written and easy to follow. There are a few questions to be clarified:

1. does the synthetic samples generated from optimizing the local objective function degenerates to a single solution? i.e, if you use IPC=10, are there any guarantee that the synthetic images are not similar to each other?

2. You use SGD in as the global optimizer, how does the method work with momentum-based optimizers? For example, if Adam is used to to update the global model.

3. In section 4.5 under "Comparison with transmitting real images", the authors claimed that optimizing on the learned synthetic dataset converges faster than optimizing on the real-data set. Transmitting the original image and update the global model on it is equivalent to a large batch SGD. The authors claimed that this is inferior to SGD on the synthetic data. I do not understand why this would be the case. More explanation and experiments on this are desired.

In addition, there are a few paper on synthetic data generation in Federated Learning, that takes very similar approaches to this paper:

Federated Learning via Synthetic Data
https://arxiv.org/abs/2008.04489

FedSynth: Gradient Compression via Synthetic Data in Federated Learning
https://arxiv.org/abs/2204.01273

---

### Official Review · Reviewer_D3oc · 2022-10-16
**Review for FedDM**

This paper considers a central issue of the non-i.i.d. federated learning scenario, distribution matching. To my knowledge, a new iterative model average algorithm is proposed based on the reconstruction of local losses by distribution matching. This method is able to reduce the communication overhead and improve the testing accuracy. The paper is well written. Multiple numerical experiments are performed based on the real datasets. It turns out the proposed method outperforms the others. One minor concern is that there is no sufficient theoretical analysis for the model and algorithms. This is a gap between the Th.D1/D2 and the proposed model. Convergence of the algorithm is also not discussed in depth.

---

### Official Review · Reviewer_Dgab · 2022-10-18

Paper summary
=============
The author proposes an FL algorithm called Federated Distribution Matching (FedDM) that uses synthetic datasets to update the global model. The paper claims that it reduces communication rounds and improves model quality compared to traditional FL algorithms such as FedAvg by transmitting condensed information, and can be adapted to preserve the same level of differential privacy with better model quality.


Pros
====
1. The method for works for extremely skewed data drawn from $Dir_{10}(0.01)$. Having an algorithm that works with non-i.i.d data is essential for its practicability.

2. The author compared well-known algorithms such as FedAvg, FedNova, and SCAFFOLD.

3. Extensive hyperparameter study is conducted to help better understand the nature of FedDM.


Cons
====
1. The Accuracy -- Comm rounds comparisons in section 3.4 and Appendix G seem unfair.

    1) In Appendix F the author states that the local iteration for FedDM is T=1000 and batch size is 256, which is roughly 5.12 epochs for CIFAR10. But the number of local epochs for SGD is chosen from [10, 15, 20].

    2) The wall time comparison is not given. Only comparing the communication rounds and communication data is not sufficient since the same communication rounds do not mean the same execution time. FedDM requires extra time for synthesizing the dataset thus analyzing the wall time overhead is essential.

2. There is no comparison for all the algorithms to be trained until convergence, especially for the CIFAR10 and CIFAR100 experiments, where FedAvg-based algorithms seem far from convergence. The author claimed improved model quality given by FedDM, but it cannot be proved if no such comparison is given.

3. The analysis using the 1-D example in Section 2.1 is invalid since the tangent line only represents the optimization direction of a single time stamp, and it changes constantly over time, thus not fitting the original curve does not necessarily mean that the model converges slower or won't converge. Moreover, we have various tricks such as momentum optimization to accelerate the convergence. Beyond that, this paper lacks solid theoretical or empirical analysis on why using synthetic is better than directly using the gradient.


Detailed comments
=================
1. In Appendix G.3 the author states that the image class is illegible in figure 14, But at least some cars/planes/dogs can be distinguished in the image...

2. Although stated in the limitations, the practical applicability and scalability of the method are still questionable. As the author stated in Section 2.2, the upload parameter size is linear to the product of the number of clients, the number of classes per client, and the image size. While Table 2 shows that for 10 clients with skewness of Dir(0.5) the uplink parameter size already exceeds the model size. Real-world scenarios often have much more classes and clients, which may deplete resources quickly.

3. The applicability to other types of tasks, such as text-based tasks or other non-classification tasks is can be explored.

---

### Decision · Program_Chairs · 2022-10-20

Accept (Poster)